# Towards Data Distillation for End-to-end Spoken Conversational Question Answering

## Abstract

In spoken question answering, QA systems are designed to answer questions from contiguous text spans within the related speech transcripts. However, the most natural way that human seek or test their knowledge is via human conversations. Therefore, we propose a new **S**poken **C**onversational **Q**uestion **A**nswering task (SCQA), aiming at enabling QA systems to model complex dialogues flow given the speech utterances and text corpora. In this task, our main objective is to build a QA system to deal with conversational questions both in spoken and text forms, and to explore the plausibility of providing more cues in spoken documents with systems in information gathering. To this end, instead of adopting automatically generated speech transcripts with highly noisy data, we propose a novel unified data distillation approach, *DDNet*, which directly fuse audio-text features to reduce the misalignment between automatic speech recognition hypotheses and the reference transcriptions. In addition, to evaluate the capacity of QA systems in a dialogue-style interaction, we assemble a **Spoken Co**nversational **Q**uestion **A**nswering (Spoken-CoQA) dataset with more than 120k question-answer pairs. Experiments demonstrate that our proposed method achieves superior performance in spoken conversational question answering.

## 1 Introduction

Conversational Machine Reading Comprehension (CMRC) has been studied extensively over the past few years within the natural language processing (NLP) communities (Zhu et al., 2018; Liu et al., 2019; Yang et al., 2019). Different from traditional MRC tasks, CMRC aims to enable models to learn the representation of the context paragraph and multi-turn dialogues. Existing methods to the conversational question answering (QA) tasks (Huang et al., 2018a; Devlin et al., 2018; Xu et al., 2019; Gong et al., 2020) have achieved superior performances on several benchmark datasets, such as QuAC (Choi et al., 2018) and CoQA (Elgohary et al., 2018). However, few studies have investigated CMRC in both spoken content and text documents.

To incorporate spoken content into machine comprehension, there are few public datasets that evaluate the effectiveness of the model in spoken question answering (SQA) scenarios. TOEFL listening comprehension (Tseng et al., 2016) is one of the related corpus for this task, an English test designed to evaluate the English language proficiency of non-native speakers. But the multi-choice question answering setting and its scale is limited to train for robust SCQA models. The rest two spoken question answering datasets are Spoken-SQuAD (Li et al., 2018) and ODSQA (Lee et al., 2018), respectively. However, there is usually no connection between a series of questions and answers within the same spoken passage among these datasets. More importantly, the most common way people seek or test their knowledge is via human conversations, which capture and maintain the common ground in spoken and text context from the dialogue flow. There are many real-world applications related to SCQA tasks, such as voice assistant and chat robot.

In recent years, neural network based methods have achieved promising progress in speech processing domain. Most existing works first select a feature extractor (Gao et al., 2019), and then enroll the feature embedding into the state-of-the-art learning framework, as used in single-turn spoken language processing tasks such as speech retrieval (Lee et al., 2015; Fan-Jiang et al., 2020; Karakos et al., 2020), translation (Bérard et al., 2016; Serdyuk et al., 2018; Di Gangi et al., 2020; Tu et al., 2020) and recognition (Zhang et al., 2017; Zhou et al., 2018; Bruguier et al., 2019; Siriwardhana

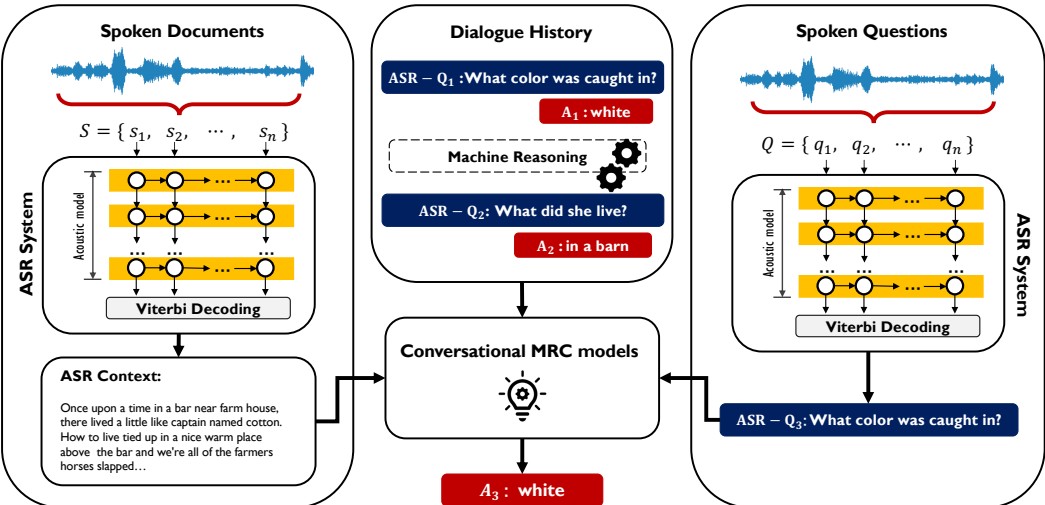

Figure 1: An illustration of flow diagram for spoken conversational question answering tasks with an example from our proposed Spoken-CoQA dataset.

Table 1: Comparison of Spoken-CoQA with existing spoken question answering datasets.

| Dataset | Conversational | Spoken | Answer Type |
|---------|:---:|:---:|:---:|
| TOEFL (Tseng et al., 2016) | × | √ | Multi-choice |
| Spoken-SQuAD (Li et al., 2018) | × | √ | Spans |
| ODSQA (Lee et al., 2018) | × | √ | Spans |
| Spoken-CoQA | √ | √ | Free-form text, Unanswerable |

et al., 2020). However, simply adopting existing methods to the SCQA tasks will cause several challenges. First, transforming speech signals into ASR transcriptions is inevitably associated with ASR errors (See Table 2). Previous work (Lee et al., 2019) shows that directly feed the ASR output as the input for the following down-stream modules usually cause significant performance loss, especially in SQA tasks. Second, speech corresponds to a multi-turn conversation (e.g. lectures, interview, meetings), thus the discourse structure will have more complex correlations between questions and answers than that of a monologue. Third, additional information, such as audio recordings, contains potentially valuable information in spoken form. Many QA systems may leverage kind of orality to generate better representations. Fourth, existing QA models are tailored for a specific (text) domain. For our SCQA tasks, it is crucial to guide the system to learn kind of orality in documents.

In this work, we propose a new spoken conversational question answering task - SCQA, and introduce Spoken-CoQA, a spoken conversational question answering dataset to evaluate a QA system whether necessary to tackle the task of question answering on noisy speech transcripts and text document. We compare Spoken-CoQA with existing SQA datasets (See Table 1). Unlike existing SQA datasets, Spoken-CoQA is a multi-turn conversational SQA dataset, which is more challenging than single-turn benchmarks. First, every question is dependent on the conversation history in the Spoken-CoQA dataset. It is thus difficult for the machine to parse. Second, errors in ASR modules also degrade the performance of machines in tackling contextual understanding with context paragraph. To mitigate the effects of speech recognition errors, we then present a novel knowledge distillation (KD) method for spoken conversational question answering tasks. Our first intuition is speech utterances and text contents share the dual nature property, and we can take advantage of this property to learn these two forms of the correspondences. We enroll this knowledge into the *student* model, and then guide the *student* to unveil the bottleneck in noisy ASR outputs to boost performance. Empirical results show that our proposed *DDNet* achieves remarkable performance gains in SCQA tasks. To the best of our knowledge, we are the first work in spoken conversational machine reading comprehension tasks.

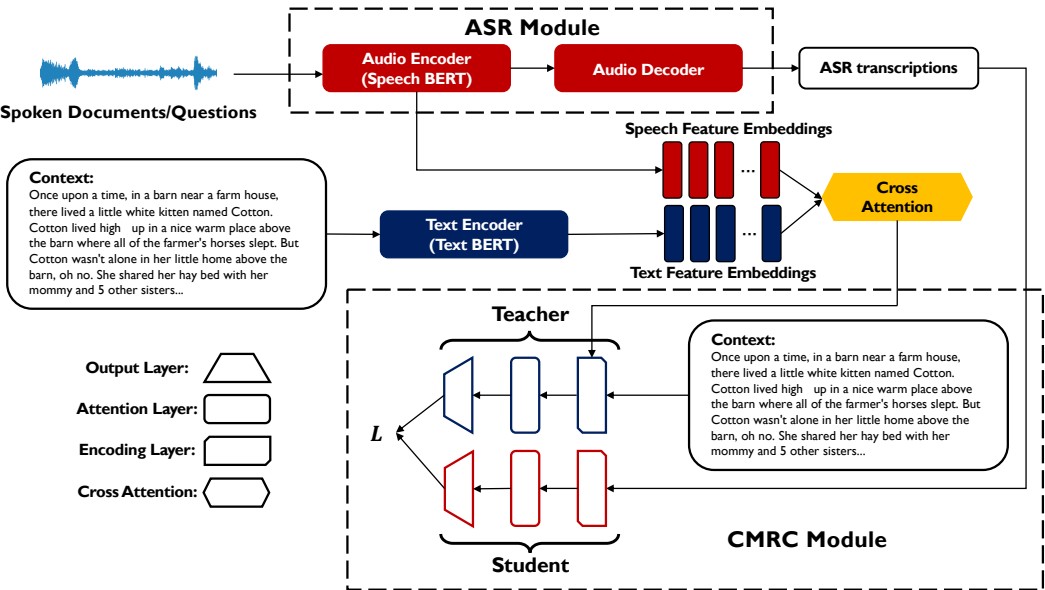

Figure 2: An illustration of the architecture of *DDNet*.

In summary, the main contributions of this work are as follows:

- We propose a new task for machine comprehension of spoken question-answering style conversation to improve the network performance. To the best of our knowledge, our Spoken-CoQA is the first spoken conversational machine reading comprehension dataset.

- We develop a novel end-to-end method based on data distillation to learn both from speech and text domain. Specifically, we use the model trained on clear syntax and close-distance recording to guide the model trained on noisy ASR transcriptions to achieve substantial performance gains in prediction accuracy.

- We demonstrate the robustness of our *DDNet* on Spoken-CoQA, and demonstrates that the model can effectively alleviate ASR errors in noisy conditions.

## 2 RELATED WORK

**Conversational Machine Reading Comprehension**  In recent years, the natural language processing research community has devoted substantial efforts to conversational machine reading comprehension tasks (Huang et al., 2018a; Zhu et al., 2018; Xu et al., 2019; Zhang et al., 2020; Gong et al., 2020). Within the growing body of work on conversational machine reading comprehension, two signature attributes have emerged: the availability of large benchmark datasets (Choi et al., 2018; Elgohary et al., 2018; Reddy et al., 2019) and pre-trained language models (Devlin et al., 2018; Liu et al., 2019; Lan et al., 2020). However, these existing works typically focus on modeling the complicated context dependency in text form. In contrast, we focus on enabling the machine to build the capability of language recognition and dialogue modeling in both speech and text domains.

**Spoken Question Answering**  In parallel to the recent works in natural language processing, these trends have also been pronounced in the speech processing (SP) field, where spoken question answering, an extended form of Question Answering, have explored the prospect of machine comprehension in spoken form. Previous work on SQA typically includes two separate modules: automatic speech recognition and text question answering. It entails transferring spoken content to ASR transcriptions, and then employs natural language processing techniques to handle the speech language processing tasks. Prior to this point, the existing methods (Tseng et al., 2016; Serdyuk et al., 2018; Su & Fung, 2020) focus on optimizing each module in a two-stage manner, where errors in the ASR module would suffer from severe performance loss. Concurrent with our research, Serdyuk et al.

(2018) proposes an end-to-end approach for natural language understanding (NLU) tasks. Speech-BERT (Chuang et al., 2019) cascades the BERT-based models as a unified model and then trains it in an audio-and-text jointly learned manner. However, the existing SQA methods aim at solving a single question given the related passage without building and maintaining the connections of different questions within the human conversations.

**Knowledge Distillation** Hinton et al. (2015) introduces the idea of Knowledge Distillation (KD) in a *teacher-student* scenario. In other words, we can distill the knowledge from one model (massive or *teacher* model) to another (small or *student* model). Previous work has shown that KD can significantly boost prediction accuracy in natural language processing and speech processing (Kim & Rush, 2016; Hu et al., 2018; Huang et al., 2018b; Hahn & Choi, 2019), while adopting KD-based methods for SCQA tasks has been less explored. Although we share the same research topic and application, our research direction and methods differ. Previous methods design a unified model to model the single-turn speech-language task. In contrast, our model explores the prospect of handling SQA tasks. More importantly, we focus the question of nature property in speech and text: do spoken conversational dialogues can further assist the model to boost the performance. Finally, we incorporate the knowledge distillation framework to distill reliable dialogue flow from the spoken contexts, and utilize the learned predictions to guide the *student* model to train well on the noisy input data.

## 3 TASK DEFINITION

### 3.1 DATA FORMAT

We introduce Spoken-CoQA, a new spoken conversational machine reading comprehension dataset where the documents are in the spoken and text form. Given the spoken multi-turn dialogues and spoken documents, the task is to answer questions in multi-party conversations. Each example in this dataset is defined as follows: $\{D_i, Q_i, A_i\}_1^N$, where $Q_i = \{q_{i1}, q_{i2}, ..., q_{iL}\}$ and $A_i = \{a_{i1}, a_{i2}, ..., a_{iL}\}$ represent a passage with $L$-turn queries and corresponding answers, respectively. Given a passage $D_i$, multi-turn history questions $\{q_{i1}, q_{i2}, ..., q_{iL-1}\}$ and the reference answers $\{a_{i1}, a_{i2}, ..., a_{iL-1}\}$, our goal is to generate $a_{iL}$ for the given current question $q_{iL}$. In this study, we use the spoken form of questions and documents as the network input for training. Note that questions and documents (passages) in Spoken-CoQA are in both text and spoken forms, and answers are in the text form.

### 3.2 DATA COLLECTION

We detail the procedures to build Spoken-CoQA as follows. First, we select the conversational question-answering dataset CoQA (Reddy et al., 2019) since it is one of the largest public CMRC datasets. CoQA contains around 8k stories (documents) and over 120k questions with answers. The average dialogue length of CoQA is about 15 turns, and the answer is in free-form text. In CoQA, the training set and the development set contain 7,199 and 500 conversations over the given stories, respectively. Therefore, we use the CoQA training set as our reference text of the training set and the CoQA development set as the test set in Spoken-CoQA. Then we employ the Google text-to-speech system to transform questions and documents in CoQA into the spoken form. Next, we adopt CMU Sphinx to transcribe the processed spoken content into ASR transcriptions. As such, we collect more than 40G audio data, and the data duration is around 300 hours. It is worth to note that since the constructed dataset does not update the answer spans based on the noisy ASR text and continues to assume answer-spans as per the actual text, we perform data filtering in our investigation by eliminating question-answer pairs from the corpus if answer spans to questions do not exist in the referenced ASR transcriptions.

For clarity, we provide an example of our Spoken-CoQA dataset in Table 2. Figure 4 compares spectrograms of samples from ASR modules. In this example, we observe that given the text document (ASR-document), the conversation starts with the question $Q_1$ (ASR-$Q_1$), and then the system requires to answer $Q_1$ (ASR-$Q_1$) with $A_1$ based on a contiguous text span $R_1$. Compared to the existing benchmark datasets, ASR transcripts (both the document and questions) are much more difficult for the machine to comprehend questions, reason among the passage, and even predict the correct answer.

Table 2: An example from Spoken-CoQA. We can observe large misalignment between the manual transcripts and the corresponding ASR transcripts. Note that the misalignment is in **bold** font.

| Manual Transcript | ASR Transcript |
| --- | --- |
| Once upon a time, in a barn near a farm house, there lived a little white kitten named Cotton. Cotton lived high up in a nice warm place above the barn where all of the farmer's horses slept. But Cotton wasn't alone in her little home above the barn, oh no. She shared her hay bed with her mommy and 5 other sisters... | Once upon a time in a **bar** near farm house, there lived a little **like captain** named cotton. **How to live** tied up in a nice warm place above the **bar** and **we're** all of the farmers horses slapped. But cotton was not alone in her little home above the bar **in now**. She shared her hey bed with her mommy and 5 other sisters... |
| $Q_1$: Did Cotton live alone?
$A_1$: no
$R_1$: Cotton wasn't alone. | ASR-$Q_1$: Did **caught in** live alone?
$A_1$: no
$R_1$: Cotton wasn't alone. |
| $Q_2$: Who did she live with?
$A_2$: with her mommy and 5 sisters
$R_2$: with her mommy and 5 other sisters | ASR-$Q_2$: Who did she live with?
$A_2$: with her mommy and 5 sisters
$R_2$: with her mommy and 5 other sisters |
| $Q_3$:What color were her sisters?
$A_3$:orange and white
$R_3$: her sisters were all orange with beautiful white tiger stripes | ASR-$Q_3$: What color were her sisters?
$A_3$: orange and white
$R_3$: her sisters were all orange with beautiful white tiger stripes |

## 4 DDNET

In this section, we detail our data distillation approach by leveraging the dual nature of speech and text domains to boost the prediction accuracy in a spoken dialogue system. An overview pipeline of this task is shown in Figure 1. We first introduce the multi-modality fusion mechanism. Then we present the major components of the CRMC module. Finally, we describe a simple yet effective distillation strategy in the proposed *DDNet* to learn feature representation in the speech-text domain comprehensively.

Given spoken words $S = \{s_1, s_2, ..., s_n\}$ and corresponding text words $X = \{x_1, x_2, ..., x_n\}$, we utilize Speech-BERT and Text-BERT to generate speech feature embedding $E_s=\{E_{s1}, E_{s2}, ..., E_{sn}\}$ and context word embedding $E_x=\{E_{x1}, E_{x2}, ..., E_{xn}\}$, respectively. Concretely, we first use vq-wav2vec (Baevski et al., 2019) to transfer speech signals into a series of tokens, which is the standard tokenization procedure in natural language processing tasks, and then use Speech-BERT (Chuang et al., 2019), a variant of BERT-based models, to process the speech sequences for training. We retrain Speech-BERT (Chuang et al., 2019) on our Spoken-CoQA dataset. The scale of Speech-BERT is similar with BERT-base (Devlin et al., 2018) model that contains 12 transformer layers with the residual structure and the embedding dimension is with 768. In parallel, we embed the text context into a sequence of vectors via our text encoder - Text-BERT. We adopt the same architecture of BERT-base (Devlin et al., 2018) in our Text-BERT due to its superior performance.

**Cross Attention** Inspired by ViLBERT (Lu et al., 2019), we apply the co-attention transformer layer(Lu et al., 2019), a variant of Self-Attention (Vaswani et al., 2017), as the Cross Attention module for speech and text embedding fusion. We pass query, key, and value matrices ($\mathbf{Q}$, $\mathbf{K}$, $\mathbf{V}$) as input to the Cross Attention module. We then compute the cross attention-pooled features by querying one modality with $\mathbf{Q}$ vector from another modality.

$$\hat{E}_s^{cross} = CrossAttention(E_s, E_x, E_x) \tag{1}$$

$$\hat{E}_x^{cross} = CrossAttention(E_x, E_s, E_s) \tag{2}$$

Finally, we obtain the aligned cross attention embedding $E_{cross}$ by concatenating $\hat{E}_s^{cross}$ and $\hat{E}_x^{cross}$.

### 4.1 Key Components

We build our CMRC module, based on recent works (Zhu et al., 2018; Huang et al., 2017). We divide our CMRC module into three key components: Encoding Layer, Attention Layer and Output Layer.

**Encoding Layer** We encode documents and conversations (questions and answers) into the corresponding feature embedding (e.g.,character embedding, word embedding, and contextual embedding), and then concatenate the output contextual embedding and the aligned cross attention embedding $E_{cross}$, and pass it as input.

$$\hat{E}_{enc} = [E_{enc}; E_{cross}] \tag{3}$$

**Attention Layer** We compute the attention on the context representations of the documents and questions, and extensively exploit correlations between them. Note that we adopt the default attention layers in four baseline models.

**Output Layer** After obtaining attention-pooled representations, the Output Layer computes the probability distribution of the start and end index within the entire documents and predicts an answer to the current question.

### 4.2 Knowledge Distillation

For prior speech-language models, the only guidance is the standard training objective to measure the difference between the prediction and the reference answer. However, such criteria makes no sense for noisy ASR transcriptions. To tackle this issue, we distill the knowledge from our *teacher* model, and use them to guide the *student* model to learn contextual features in our spoken CMRC task. Concretely, we set the model trained on the speech document and text corpus as the *teacher* model and trained on the ASR transcripts as the *student* model, respectively. Thus, the *student* trained on low-quality data learn to imbibe the knowledge that the *teacher* has discovered.

Concretely, given the $z_S$ and $z_T$ are the prediction vectors by the *student* and *teacher* models, the objective is define as:

$$L = \sum_{x \in \mathcal{X}} (\alpha \tau^2 \mathcal{KL}(p_\tau(z_S), p_\tau(z_T)) + (1 - \alpha)\mathcal{XE}(z_T, y)), \tag{4}$$

where $\mathcal{KL}(\cdot)$ and $\mathcal{XE}(\cdot)$ denote the Kullback-Leibler divergence and cross entropy, respectively. $y$ represents the ground truth labels in the text training dataset $X$. $p_\tau(\cdot)$ refers the softmax function with temperature $\tau$, and $\alpha$ is a balancing factor.

## 5 Experiments and Results

In this section, we first introduce several state-of-the-art language models as our baselines, and then evaluate the robustness of these models on our proposed Spoken-CoQA dataset. Finally, we provide a thorough analysis of different components of our method. Note that we use the default setting in all the evaluated methods.

### 5.1 Baselines

In principle, *DDNet* can utilize any backbone network for SCQA tasks. We choose several state-of-the-art language models (FlowQA (Huang et al., 2018a), SDNet (Zhu et al., 2018), BERT-base (Devlin et al., 2018), ALBERT (Lan et al., 2020)) as our backbone network due to its superior performance. We also compare our proposed *DDNet* with several state-of-the-art SQA methods (Lee et al., 2018; Serdyuk et al., 2018; Lee et al., 2019; Kuo et al., 2020). To train the *teacher-student* pairs simultaneously, we first train baselines on the CoQA training set and then compare the performances of testing baselines on CoQA dev set and Spoken-CoQA dev set. Finally, we train the baselines on the Spoken-CoQA training set and evaluate the baselines on the CoQA dev set and Spoken-CoQA test set. We provide quantitative results in Table 3.

Table 3: Comparison of four baselines (FlowQA, SDNet, BERT, ALBERT). Note that we denote Spoken-CoQA test set as S-CoQA test for brevity.

| | CoQA | | | | S-CoQA | | | |
| | CoQA dev | | S-CoQA test | | CoQA dev | | S-CoQA test | |
| Methods | EM | F1 | EM | F1 | EM | F1 | EM | F1 |
|---|---|---|---|---|---|---|---|---|
| FlowQA (Huang et al., 2018a) | 66.8 | 75.1 | 44.1 | 56.8 | 40.9 | 51.6 | 22.1 | 34.7 |
| SDNet (Zhu et al., 2018) | 68.1 | 76.9 | 39.5 | 51.2 | 40.1 | 52.5 | 41.5 | 53.1 |
| BERT-base (Devlin et al., 2018) | 67.7 | 77.7 | 41.8 | 54.7 | 42.3 | 55.8 | 40.6 | 54.1 |
| ALBERT-base (Lan et al., 2020) | 71.4 | 80.6 | 42.6 | 54.8 | 42.7 | 56.0 | 41.4 | 55.2 |
| Average | 68.5 | 77.6 | 42 | 54.4 | 41.5 | 54.0 | 36.4 | 49.3 |

## 5.2 EXPERIMENT SETTINGS

We use the official BERT (Devlin et al., 2018) and ALBERT (Lan et al., 2020) as our starting point for training. We use BERT-base (Devlin et al., 2018) and ALBERT-base (Lan et al., 2020), which both include 12 transformer encoders, and the hidden size of each word vector is 768. BERT and ALBERT utilize BPE as the tokenizer, but FlowQA and SDNet use SpaCy (Honnibal & Montani, 2017) for tokenization. Specifically, in the case of tokens in spaCy (Honnibal & Montani, 2017) correspond to more than one BPE sub-tokens, we average the BERT embeddings of these BPE sub-tokens as the embedding for each token. To maintain the integrity of all evaluated model performance, we use standard implementations and hyper-parameters of four baselines for training. The balancing factor $\alpha$ is set to 0.9, and the temperature $\tau$ is set to 2. For evaluation, we use Exact Match (EM) and $F_1$ score to compare the model performance on the test set. Note that in this work, each baseline is trained in the local computing environment, which may results in different results compared with the ones on the CoQA leader board.

## 5.3 RESULTS

We compare several *teacher-student* pairs on CoQA and Spoken-CoQA dataset. Quantitative results are shown in Table 3. We can observe that the average F1 scores are 77.6% when training on CoQA (text document) and testing on the CoQA dev set. However, when training the models on Spoken-CoQA (ASR transcriptions) and testing on the Spoken-CoQA test set, average F1 scores are dropped to 49.3%. For FlowQA, the performance even dropped by 40.4% on F1 score. This confirms the importance of mitigating ASR errors which severely degrade the model performance in our tasks.

As shown in Table 4, it demonstrates that our proposed Cross Attention block and knowledge distillation strategy consistently boost the remarkable performance on all baselines, respectively. More importantly, our distillation strategy works particularly well. For FlowQA, our method achieves 53.7% (vs.51.6%) and 39.2% (vs.34.7%) in terms of F1 score over text document and ASR transcriptions, respectively. For SDNet, our method outperforms the baseline without distillation strategy, achieving 55.6% (vs.52.5%) and 56.7% (vs.53.1%) in terms of F1 score. For two BERT-like models (BERT-base and ALBERT-base), our methods also improve F1 scores to 58.8% (vs.55.8%) and 57.7% (vs.54.1%); 59.6% (vs.56.0%) and 58.7% (vs.55.2%), respectively. We also compare the combination of our distillation strategy and the cross attention mechanism. Our results suggest that such network notably improve prediction performance for spoken conversational question answering tasks. Such significant improvements demonstrate the effectiveness of *DDNet*.

## 6 QUANTITATIVE ANALYSIS

**Speech Feature in ASR System** To perform qualitative analysis of speech features, we visualize the log-mel spectrogram features and the mel-frequency cepstral coefficients (MFCC) feature embedding learned by *DDNet* in Figure 4. We can observe how the spectrogram features respond to different sentence examples.

**Temperature $\tau$** To study the effect of temperature $\tau$ (See Section 4.2), we conduct the additional experiments of four baselines with the standard choice of the temperature $\tau \in \{1, 2, 4, 6, 8, 10\}$. All models are trained on Spoken-CoQA dataset, and validated on the CoQA dev and Spoken-CoQA

Table 4: Comparison of key components in *DDNet*. We set the model on speech document and text corpus as the *teacher* model, and the one on the ASR transcripts as the *student* model.

| Methods | CoQA dev | | S-CoQA test | |
|---|---|---|---|---|
| | EM | F1 | EM | F1 |
| FlowQA (Huang et al., 2018a) | 40.9 | 51.6 | 22.1 | 34.7 |
| FlowQA + sub-word unit (Li et al., 2018) | 41.9 | 53.2 | 23.3 | 36.4 |
| FlowQA+ SLU (Serdyuk et al., 2018) | 41.2 | 52.0 | 22.4 | 35.0 |
| FlowQA + back-translation (Lee et al., 2018) | 40.5 | 52.1 | 22.9 | 35.8 |
| FlowQA + domain adaptation (Lee et al., 2019) | 41.7 | 53.0 | 23.4 | 36.1 |
| FlowQA + **Cross Attention** | 41.1 | 52.2 | 22.5 | 35.5 |
| FlowQA + **Knowledge Distillation** | 42.5 | 53.7 | 23.9 | 39.2 |
| FlowQA + **Cross Attention+Knowledge Distillation** | **42.9** | **54.7** | **24.9** | **41.0** |
| SDNet (Zhu et al., 2018) | 40.1 | 52.5 | 41.5 | 53.1 |
| SDNet + sub-word unit (Li et al., 2018) | 41.2 | 53.7 | 41.9 | 54.7 |
| SDNet+ SLU (Serdyuk et al., 2018) | 40.2 | 52.9 | 41.7 | 53.2 |
| SDNet + back-translation (Lee et al., 2018) | 40.5 | 53.1 | 42.4 | 54.0 |
| SDNet + domain adaptation (Lee et al., 2019) | 41.0 | 53.9 | 41.7 | 54.6 |
| SDNet + **Cross Attention** | 40.4 | 52.9 | 41.6 | 53.4 |
| SDNet + **Knowledge Distillation** | 41.7 | 55.6 | 43.6 | 56.7 |
| SDNet + **Cross Attention+Knowledge Distillation** | **42.1** | **56.6** | **44.0** | **57.7** |
| BERT-base (Devlin et al., 2018) | 42.3 | 55.8 | 40.6 | 54.1 |
| BERT-base + sub-word unit (Li et al., 2018) | 43.2 | 56.8 | 41.6 | 55.4 |
| BERT-base+ SLU (Serdyuk et al., 2018) | 42.5 | 56.1 | 41.0 | 54.6 |
| BERT-base + back-translation (Lee et al., 2018) | 42.9 | 56.5 | 41.5 | 55.2 |
| BERT-base + domain adaptation (Lee et al., 2019) | 43.1 | 57.0 | 41.7 | 55.7 |
| aeBERT (Kuo et al., 2020) | 43.0 | 56.9 | 41.8 | 55.6 |
| BERT-base + **Cross Attention** | 42.4 | 56.3 | 40.9 | 54.5 |
| BERT-base + **Knowledge Distillation** | 44.1 | 58.8 | 42.8 | 57.7 |
| BERT-base + **Cross Attention+Knowledge Distillation** | **44.2** | **59.8** | **43.5** | **58.4** |
| ALBERT-base (Lan et al., 2020) | 42.7 | 56.0 | 41.4 | 55.2 |
| ALBERT-base + sub-word unit (Li et al., 2018) | 43.7 | 57.2 | 42.6 | 56.8 |
| ALBERT-base + SLU (Serdyuk et al., 2018) | 42.8 | 56.3 | 41.7 | 55.7 |
| ALBERT-base + back-translation (Lee et al., 2018) | 43.5 | 57.1 | 42.4 | 56.4 |
| ALBERT-base + domain adaptation (Lee et al., 2019) | 43.5 | 57.0 | 42.7 | 56.7 |
| ALBERT-base + **Cross Attention** | 42.9 | 56.4 | 41.6 | 55.9 |
| ALBERT-base + **Knowledge Distillation** | 44.8 | 59.6 | 43.9 | 58.7 |
| ALBERT-base + **Cross Attention+ Knowledge Distillation** | **45.2** | **60.2** | **44.4** | **60.1** |

test set, respectively. In Figure 3, when $T$ is set to 2, four baselines all achieve their best performance in term of F1 and EM metrics.

**Multi-Modality Fusion Mechanism** To study the effect of different modality fusion mechanisms, we introduce a novel fusion mechanism *Con Fusion*: first, we directly concatenate two output embedding from speech-BERT and text-BERT models, and then pass it to the encoding layer in the following CMRC module. In Table 5, we observe that Cross Attention fusion mechanism outperform four baselines with *Con Fusion* in terms of EM and F1 scores. We further investigate the effect of uni-model input. Table 5 shows that *text-only* performs better than *speech-only*. One possible reason for this performance is that only using speech features can bring additional noise. Note that speech-only (text-only) represents that we only feed the speech (text) embedding for speech-BERT (text-BERT) to the encoding layer in the CMRC module.

## 7    CONCLUSION

In this paper, we have presented a new spoken conversational question answering task - Spoken-CoQA, for enabling human-machine communication. Unlike the existing Spoken conversational machine reading comprehension datasets, Spoken-CoQA includes multi-turn conversations and passages in both text and speech form. Furthermore, we propose a data distillation method, which

Table 5: Comparison of different fusion mechanisms in *DDNet*. We set the model trained on speech document and text corpus as the *teacher* model, and the one trained on the ASR transcripts as the *student* model.

| Models | CoQA dev | | S-CoQA test | |
|---|---|---|---|---|
| | EM | F1 | EM | F1 |
| FlowQA (Huang et al., 2018a) | 40.9 | 51.6 | 22.1 | 34.7 |
| FlowQA (Huang et al., 2018a)+ *speech-only* | 40.8 | 51.2 | 21.8 | 34.0 |
| FlowQA (Huang et al., 2018a)+ *text-only* | 41.1 | 51.7 | 22.4 | 35.3 |
| FlowQA (Huang et al., 2018a)+ *Con Fusion* | 41.0 | 52.0 | 22.1 | 35.2 |
| FlowQA (Huang et al., 2018a)+ **Cross Attention** | 41.1 | 52.2 | 22.5 | 35.5 |
| SDNet (Zhu et al., 2018) | 40.1 | 52.5 | 41.5 | 53.1 |
| SDNet (Zhu et al., 2018)+ *speech-only* | 39.3 | 51.6 | 40.9 | 52.28 |
| SDNet (Zhu et al., 2018)+ *text-only* | 40.2 | 52.7 | 41.5 | 53.3 |
| SDNet (Zhu et al., 2018)+ *Con Fusion* | 40.3 | 52.6 | 41.5 | 53.2 |
| SDNet (Zhu et al., 2018)+ **Cross Attention** | 40.4 | 52.9 | 41.6 | 53.4 |
| BERT-base (Devlin et al., 2018) | 42.3 | 55.8 | 40.6 | 54.1 |
| BERT-base (Devlin et al., 2018)+ *speech-only* | 41.9 | 55.8 | 40.2 | 54.1 |
| BERT-base (Devlin et al., 2018)+ *text-only* | 42.4 | 56.0 | 40.9 | 54.3 |
| BERT-base (Devlin et al., 2018)+ *Con Fusion* | 42.3 | 56.0 | 40.8 | 54.1 |
| BERT-base (Devlin et al., 2018)+ **Cross Attention** | 42.4 | 56.3 | 40.9 | 54.5 |
| ALBERT-base (Lan et al., 2020) | 42.7 | 56.0 | 41.4 | 55.2 |
| ALBERT-base (Lan et al., 2020)+ *speech-only* | 41.8 | 55.9 | 41.1 | 54.8 |
| ALBERT-base (Lan et al., 2020)+ *text-only* | 42.9 | 56.3 | 41.4 | 55.7 |
| ALBERT-base (Lan et al., 2020)+ *Con Fusion* | 42.7 | 56.1 | 41.3 | 55.4 |
| ALBERT-base (Lan et al., 2020)+ **Cross Attention** | 42.9 | 56.4 | 41.6 | 55.9 |

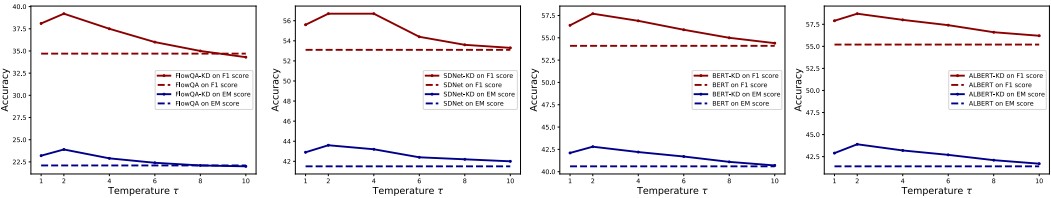

Figure 3: Ablation studies of temperature $\tau$ on *DDNet* performance (FlowQA, SDNet, BERT, AL-BERT). Red and blue denote the results on CoQA dev and Spoken-CoQA test set, respectively.

leverages audio-text features to reduce the misalignment between ASR hypotheses and the reference transcriptions. Experimental results show that *DDNet* achieves superior performance in prediction accuracy. For future work, we will further investigate different mechanism of integrating speech and text content, and propose novel machine learning based networks to migrate ASR recognition errors to boost the performance of QA systems.

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

# A  APPENDIX

## A.1  SPEECH FEATURES IN ASR SYSTEM

Due to the page limit, we present some examples of speech features here.

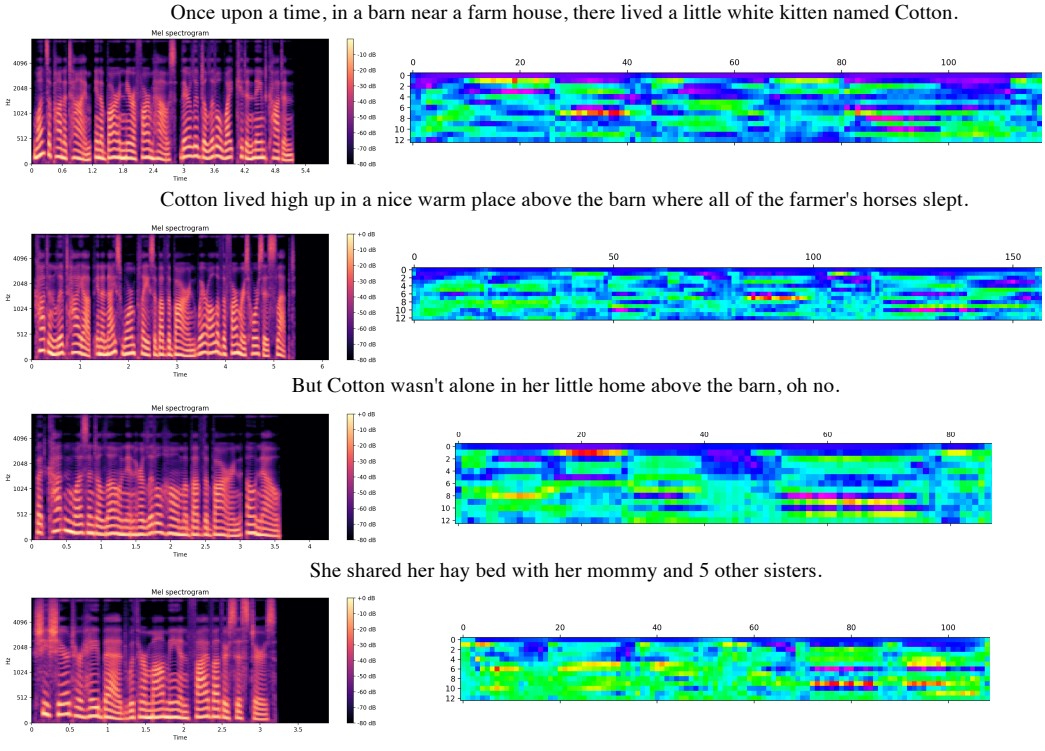

Figure 4: Examples of the log-mel spectrograms and the corresponding MFCC feature embedding. It can see that the log-mel spectrograms corresponds to different example sentences from the Spoken-CoQA dataset.

