# OpenReview forum: "Towards Data Distillation for End-to-end Spoken Conversational Question Answering"
_ICLR.cc/2021/Conference — Reject_

### Official Review · AnonReviewer3 · 2020-10-26
**Unfortunately falls short of delivering a usable dataset for spoken conversation QA**

**Rating:** 4
**Confidence:** 4

**Review:**

---------------
Summary
---------------

In this paper, the authors release a new dataset -  Spoken-CoQA which includes an ASR based version of the popular CoQA dataset. The dataset has been created by running the Google TTS system followed by ASR using CMU Sphinx, to create a speech-transcribed versions of the dataset. The dataset includes the corresponding TTS audio recordings. Since the transcribed dataset has transcription errors, existing reading comprehension models do not work well. Thus, the paper introduces a joint audio-textual model for QA on the Spoken-CoQA dataset that uses TTS recordings its corresponding ASR output.

The model encodes audio data using the recent Speech-BERT model while the text is encoded using regular BERT. Similar to visio-linguistic models such as VilBERT, the model uses cross-attention on the intermediate audio and textual representations. This is done by using query matrix of a transformer block from one modality and applying it for attention using the key and value matrices on the other. That is, query keys from text are used to attend over keys and values from audio and vice-versa. The cross-attended representations are concatenated and used as input to any "CMRC" module. The authors experiment with two "CMRC" modules - SDNet and FlowQA. These are existing models developed for QA on CoQA, to return span-extracted answers from the input passage. Note that the dataset constructed does not update the answer spans based on the noisy ASR text and continues to assume answer-spans as per the actual text. Thus, to address this the authors include a knowledge distillation (KD) layer where the teacher network uses the gold speech transcriptions with audio, while the student layer uses the noisy transcriptions.

The experiments have been presented using SDNeT, FlowQA as the CMRC modules and also by using BERT and ALBERT as the alternative CMRC modules. Models have been trained and cross-tested using CoQA and Spoken-CoQA. As expected, configurations with cross-testing report a deterioration in performance. For models trained using Spoken-CoQA, all models benefit from using audio-textual cross-attention as well as KD.

--------------------------------------
Strengths and Weaknesses
---------------------------------------
The paper is interesting overall - while the model by itself is a trivial combination of existing multi-model methods, they result in upto 2pt improvements. I'd be willing to overlook this aspect, but to me the biggest weakness of this paper is in its data construction. It appears, when the ASR output is noisy -- the spans refer to ghost token positions (based on the clean text) -- example in Table 2 where the first question is unanswerable in span positions since the word "white" does not exist in the ASR text. The authors also allude to this when they motivate the KD layer, but my concern is what does one learn with this data? Any existing CMRC model trained on this data is obviously going to be bad -- it cant learn anything meaningful. Similarly, even when one employs the audio-textual model why should the model learn to predict the wrong span? I guess that is why the models also dont improve much - with the use of audio-textual encoding and KD. I'm not sure what can be done about this -- perhaps instead of using the text-based spans, one could return the audio segments  as answers? That might be more meaningful but will require annotation. In fact doing so may also remove the need of the KD layer which in some ways tries to (incorrectly) fix the problem by also showing it the gold clean transcription (original passage) . However, I'd argue that  relying on the clean text kind of defeats the purpose of speech-based conversational QA as motivated in this paper.

Speech based conversation QA is an important problem and the authors make a good first attempt, but unfortunately the paper falls short of delivering a usable dataset for this task.

---

> ### Author Response · Authors · 2020-11-15
> **Response to Reviewer 3**
>
> Thank you for your review. We understand your sentiment and found that the comments have some significant misread, which we want to point out right away:
>  - The reviewer mentioned, quote *“ Note that the dataset constructed does not update the answer spans based on the noisy ASR text and continues to assume answer-spans as per the actual text.”* We very appreciate you point out this, but in fact we have done the data filtering in our investigation as follows: If the answer spans to questions do not exist in the referenced ASR transcriptions, we removed question-answer pairs from our Spoken-CoQA dataset because these examples are too complicated for spoken machine comprehension. Finally, we collected more than 40k question-answer pairs. As a result, we believe it will alleviate the reviewer's concerns, such as mentioned *“it cannot learn anything meaningful.”* And we feel that this is critical to the understanding, so we have added a detailed description in the manuscript.
>  - We will release our dataset and all the codes after the publication.
>  - Regarding the comment, quote *“relying on the clean text kind of defeats the purpose of speech-based conversational QA”.* Inspired by the recent work [1,2], we use clean text as an auxiliary tool to solve this speech-based conversational question answering task.
>
> [1] Spoken SQuAD: A Study of Mitigating the Impact of Speech Recognition Errors on Listening Comprehension
>
> [2] ODSQA: Open-domain Spoken Question Answering Dataset

---

### Official Review · AnonReviewer2 · 2020-10-29
**Good paper, could be better**

**Rating:** 5
**Confidence:** 3

**Review:**

This paper studies spoken conversational QA and its main contributions includes:

- It proposes a task for end-to-end spoken QA. The new dataset Spoken-CoQA is derived from CoQA with additional features including audio data and ASR transcripts.

- It proposes a method utilizing data distillation to learn from speech and text jointly.

The paper is technically sound and well structured. However, maybe I missed something, but I am not entirely sure if the contributions and its novelty warrant an accept.

While the two methods, (1) the cross attention mechanism for speech and text embedding fusion and (2) knowledge distillation for combatting ASR errors, both bring improvement to the base models, they are existing, well-studied methods. The most important contribution of this paper, in my opinion, is the construction of the Spoken-CoQA dataset. But it seems the dataset isn't made available.

Between knowledge distillation and cross attention, it seems the former brings a larger improvement while the latter generally only has a marginal effect (Table 4). On this result, I have two questions: (1) Can these two mechanisms be combined (to reach an even better performance)? (2) Is it correct to say that the textual input is more useful than the audio input in this dataset? This might be an interesting question especially as the audio input is much larger and hence more difficult to process.

---

> ### Author Response · Authors · 2020-11-15
> **Response to Reviewer 2**
>
> Thanks for the review. We found that the comments have some significant misread, which we want to point out right away:
>  - Regarding the comment, quote *“not entirely sure if the contributions and its novelty warrant an accept.”*, we want to point out that our work is the first attempt to propose a new task - **spoken conversational question answering**, and there exist many applications related to this in the real life. Furthermore, we release a new dataset - Spoken-CoQA, and present a novel end-to-end distillation model for the spoken conversational question answering task.
>  - Regarding the comment, quote *“it seems the dataset isn't made available.”*,  we will release our dataset and all the codes after the publication.
>  - Regarding the comment, quote *“Can these two mechanisms be combined (to reach an even better performance)?”* We sincerely appreciate the reviewer to point out this. Actually, we have done the combination work. However, since we want to demonstrate the effectiveness of the distillation strategy, we report the results of cross attention and knowledge distillation in Table 4, separately. We have added the results of a combination of two mechanisms in Table 4. Our results demonstrate that the combination of two mechanisms can bring significant performance improvements. Meanwhile, we compared several state-of-the-art spoken question answering methods [1,2,3] in Table 4 of the revised manuscript. This suggests that the proposed DDNet achieves remarkable performance on SCQA tasks.
>  - Regarding the comment, quote *"Is it correct to say that the textual input is more useful than the audio input in this dataset?”*, we present the results in Table 5 in the Appendix. As shown in the Table, we argue that it is hard to quantify the importance of textual input or audio input for the spoken conversational question answering (SCQA) task. Our main goal is to find an effective way to address the SCQA task. And some previous works [4,5] may shed light on this.
>
> [1] Spoken SQuAD: A Study of Mitigating the Impact of Speech Recognition Errors on Listening Comprehension
>
> [2] Mitigating the Impact of Speech Recognition Errors on Spoken Question Answering by Adversarial Domain Adaptation.
>
> [3] ODSQA: Open-domain Spoken Question Answering Dataset
>
> [4] SpeechBERT: An Audio-and-text Jointly Learned Language Model for End-to-end Spoken Question Answering
>
> [5] An Audio-enriched BERT-based Framework for Spoken Multiple-choice Question Answering

---

### Official Review · AnonReviewer4 · 2020-10-29
**Propose a spoken conversational question answering system**

**Rating:** 5
**Confidence:** 4

**Review:**

The authors tackle the problem of spoken conversational question answering involving multiple turns of a dialogue, where the documents and questions are both in spoken and text form. The authors also compile a new dataset for spoken conversational QA with the help of text-to-speech systems.

In its current form, I think this work is fairly limited in its scope and is not yet ready to be published at ICLR. Other than ablations of the proposed technique and different choices of backbone networks for the proposed DDNet framework, since there are no comparisons made to prior approaches it is hard to assess the merits of the proposed approach. The authors also do not mention any plans of releasing the new dataset described in this work.

The draft will also benefit from a thorough editing pass; there are many typos (e.g., "tailed for a specific domain", "nature language processing", "BRET-base", etc).

Three other comments:
* It might be interesting to show how F1 scores vary on the test instances as a function of the number of turns in the conversation. Do the ASR errors compound to hurt performance on conversations with a large number of turns or is that not much of an issue? Similarly, showing how test F1 scores vary as a function of ASR accuracy of the spoken documents/questions would also be interestin
g.
* What is the error rate of the ASR system on the spoken documents and spoken questions? This will give the reader an idea of the accuracy of the transcriptions fed as input to the student model.
* In Table 4, the F1 scores using SDNet are higher for S-CoQA compared to CoQA which is unexpected. Could the authors comment on why this might be?

------------

Update after author response:

I've increased my score to 5. However, I still think this work is not ready to be published at ICLR in its current form. One of the other reviewers had raised an important point about the reliance of the proposed system on clean text which the authors should consider addressing in an updated version of this work.

---

> ### Author Response · Authors · 2020-11-15
> **Response to Reviewer 4**
>
> We thank the reviewer for the detailed questions. Please see our responses below, and we would appreciate follow up discussions if anything remains unclear.
>  - To our best knowledge, we are the first attempt for spoken conversational question answering task. There is no specific baseline for this task. We sincerely appreciate the reviewer to point out this. We compared several state-of-the-art spoken question answering methods [1,2,3] in Table 4 of the revised manuscript. Our results suggest that the proposed DDNet achieves remarkable performance on SCQA tasks.
>  - We will release our dataset and all the codes after the publication.
>  - Sorry for the typos. We have revised in the manuscripts.
>  - As mentioned, quote *“Do the ASR errors compound to hurt performance on conversations with a large number of turns or is that not much of an issue? “*  We have observed this phenomenon and experiments are underway. As for the comment, quote *“Similarly, showing how test F1 scores vary as a function of ASR accuracy of the spoken documents/questions would also be interesting”*, our research mainly focuses on the distillation approach to directly fuse audio-text features to reduce the misalignment between automatic speech recognition hypotheses and the reference transcriptions. We will investigate this in our future work.
>  - The word error rate of the ASR system is 17.3% on the spoken documents and spoken questions in our investigation.
>  - As mentioned, quote *“the F1 scores using SDNet are higher for S-CoQA compared to CoQA”*, the main possible reason is SDNet employs BERT as its encoders since the latter is trained on a large benchmark dataset with clean corpus, which may obtain the performance improvements to ASR transcriptions.
>
> [1] Spoken SQuAD: A Study of Mitigating the Impact of Speech Recognition Errors on Listening Comprehension
>
> [2] Mitigating the Impact of Speech Recognition Errors on Spoken Question Answering by Adversarial Domain Adaptation.
>
> [3] ODSQA: Open-domain Spoken Question Answering Dataset

---

### Official Review · AnonReviewer1 · 2020-10-30

**Rating:** 6
**Confidence:** 3

**Review:**

This paper proposes a new task: spoken conversational question answering, which combines conversational question answering (e.g. CoQA) with spoken question answering (e.g. Spoken-SQuAD). The task is to answer a question (in written text) given a question that is given in both audio form and text form. They create a dataset for this task by combining CoQA with some off-the-shelf text-to-speech and speech-to-text models. They then propose a new model, DDNet, which obtains improved performance on their dataset.

Pros:

-I buy the general argument made in the paper that relying solely on transcribed text from audio for answering the questions could be problematic, so I can see the value in the proposed dataset. I also thought the way the dataset is produced, by running CoQA through a text-to-speech model, was fairly clever. I’m not very familiar with the literature on spoken question answering to know if this is a common practice.

-Another positive of the paper is that the DDNet architecture does seem to improve the performance on their dataset by a fairly large amount (though no error bars are given).

-I appreciate that the paper conducts ablations on the different model components.

Cons:
-In my view, a drawback of this paper is that it is a bit difficult to read (partially due to grammatical errors). I found much of the motivation for this new task to not be clear from reading the paper. For example, I found the following excerpt difficult to parse:
“Unlike existing SQA datasets, Spoken-CoQA is a multi-turn conversational SQA dataset, which is more challenging than single-turn benchmarks. First, every question is dependent on the conversation history in the Spoken-CoQA dataset. It is thus difficult for the machine to parse. Second, errors in ASR modules also degrade the performance of machines in tackling contextual understanding with context paragraph.“

-Similarly, I found Figure 1 to be very confusing.


-As far as I can tell, the paper doesn't compare to any other baselines that incorporate the audio information in a different way than DDNet. For example, the method from Serdyuk et al. (2018) could be considered. Strangely, the paper claims that this paper was ‘concurrent’, despite the fact that it was published in 2018, which is very confusing to me.

-It’s unclear why the Spoken-CoQA dataset has to include text transcripts as well as the audio --- to me, it makes more sense for that to be part of the model solving the dataset.

- Ideally the dataset would have natural speech, instead of synthetic speech, but I don’t consider this a major limitation.

Overall:
I think this is a borderline paper, erring on the side of rejection. My main concern is the lack of audio-based baselines other than DDNet, and the clarity of the paper.

******* Update after reading rebuttal ********
I appreciate the additional comparisons to prior work added by the authors. I've raised my score to a 6, but I still view this as a borderline paper due to concerns about clarity and impact, along with the  concerns raised by Reviewer 3.

---

> ### Author Response · Authors · 2020-11-15
> **Response to Reviewer 1**
>
> Thanks for the review. Below we want to address several questions that we think are key points to clarify first:
>  - We found out that we had a proofreading mistake in Figure 1, where the $A_2$ should be $A_3$. We have updated it in the revised manuscript.
>  - To our best knowledge, we are the first attempt for spoken conversational question answering task. There is no specific baseline for this task. We have conducted a comparison with the method proposed by Serdyuk et al. (2018) (See Table 4). The method is designed to address the spoken language understanding (SLU) task. In comparison, our spoken conversational question answering task includes spoken/natural language understanding, and conversational question answering, where is much more challenging than the SLU task. Furthermore, our proposed *DDNet* shares a similar network architecture but has a more considerable amount of parameters than the method proposed by Serdyuk et al. (2018). We sincerely appreciate the reviewer to point out this. We compared several state-of-the-art spoken question answering methods [1,2,3] in Table 4 of the revised manuscript. Our results suggest that the proposed *DDNet* achieves remarkable performance on SCQA tasks.
>  - We agree that the Spoken-CoQA dataset is necessary to include text transcripts as well as the audio. However, inspired by the recent success (Spoken-SQuAD [1] and ODSQA [2]), we aim at leveraging text transcripts to improve model performance. As shown in Tables 4 and 5, it demonstrates that the text-audio features indeed enable the model to achieve better performance compared with the same model trained on the audio-only Spoken-CoQA dataset.
>
> [1] Spoken SQuAD: A Study of Mitigating the Impact of Speech Recognition Errors on Listening Comprehension
>
> [2] Mitigating the Impact of Speech Recognition Errors on Spoken Question Answering by Adversarial Domain Adaptation.
>
> [3] ODSQA: Open-domain Spoken Question Answering Dataset

---

### Decision · Program_Chairs · 2021-01-07
**Final Decision**

**Decision:**

Reject

**Comment:**

The authors propose a dataset and a method for the task of SpokenQA. The dataset is generated by using Google TTS to generate audio segments corresponding to the CoQA dataset and then using an ASR system to generate (noisy) transcripts of these speech segments. The authors then propose a method which uses a combination of various known techniques.

Pros:
- A good first attempt at creating an interesting dataset for a useful task

Cons:
- Lack of clarity in writing
- Use of original clean text as input to the model which beats the purpose (in a natural setting such clean text will not be available)

All reviewers have appreciated the effort and attempt at creating a new dataset for this task. However, they have also pointed out that paper is not very clearly written and some important issues (use of clean text as input to the model) need to be adequately addressed before the paper is ready for acceptance.